# Linear Dynamic Viscoelasticity of Dual Cross-Link Poly(Vinyl Alcohol) Hydrogel with Determined Borate Ion Concentration

**DOI:** 10.3390/gels7020071

**Published:** 2021-06-14

**Authors:** Takuro Taniguchi, Kenji Urayama

**Affiliations:** Department of Macromolecular Science and Engineering, Kyoto Institute of Technology, Matsugasaki, Sakyo-ku, Kyoto 606-8585, Japan; taniguchipolymer@gmail.com

**Keywords:** polymer hydrogel, temporary network, transient network, viscoelasticity, rheology

## Abstract

We investigated the linear dynamic viscoelasticity of dual cross-link (DC) poly(vinyl alcohol) (PVA) (DC-PVA) hydrogels with permanent and transient cross-links. The concentrations of incorporated borate ions to form transient cross-links in the DC-PVA hydrogels (*C*_B_^IN^) were determined by the azomethine-H method. The dynamic viscoelasticity of the DC-PVA hydrogel cannot be described by a simple sum of the dynamic viscoelasticity of the PVA gel with the same permanent cross-link concentration and the PVA aqueous solution with the same borate ion concentration (*C*_B_ = *C*_B_^IN^) as in the DC-PVA gel. The DC-PVA hydrogel exhibited a considerably higher relaxation strength, indicating that the introduction of permanent cross-links into temporary networks increases the number of viscoelastic chains with finite relaxation times. In contrast, the relaxation frequency (*ω*_c_) (given by the frequency at the maximum of loss modulus) for the DC-PVA hydrogel was slightly lower but comparable to that for a dilute PVA solution with the same *C*_B_. This signifies that the relaxation dynamics of the DC-PVA hydrogels is essentially governed by the lifetime of an interchain transient cross-link (di-diol complex of boron). The effect of permanent cross-linking on the relaxation dynamics was observed in the finite broadening of the relaxation-time distribution in the long time region.

## 1. Introduction

Hydrogels have substantial potential for application in industrial and biomedical fields owing to their unique characteristics such as low modulus, high extensibility, low surface friction, and swelling properties [1,2]. The mechanical toughening of hydrogels has been gaining considerable attention because hydrogels with high toughness are crucial for industrial applications [3]. Most hydrogels with permanent cross-links formed by covalent bonds do not have sufficiently high toughness. The introduction of transient (physical) cross-links with a finite characteristic time into hydrogels significantly enhances the mechanical toughness of the hydrogels [4,5,6,7,8]. The incorporation of transient cross-links provides permanent networks with an energy dissipation mechanism resulting from the viscoelastic relaxation of the temporary chains with both ends (or one end) connected to transient cross-links.

Narita et al. [7]. reported the preparation of poly(vinyl alcohol) (PVA) hydrogels with permanent and transient cross-links, termed “dual cross-link” (DC) hydrogels (Figure 1). The permanent cross-link is formed by a covalent bond between glutaraldehyde (GA) and OH groups, whereas the transient cross-link is established by a dynamic covalent bond via association/dissociation between a tetrahydroxy borate ion and OH groups. The addition of borate ions to an aqueous PVA solution substantially increases the solution viscosity owing to the formation of the transient cross-links [9,10,11,12,13]. Although PVA solutions with physical cross-links formed by borate ions (P-PVA solutions) are considerably viscoelastic, they flow over a long period of time. PVA hydrogels with only permanent chemical cross-links (C-PVA hydrogels) are elastic solids; however, they exhibit almost no energy dissipation. In contrast, a DC-PVA hydrogel behaves as a viscoelastic solid with an energy dissipation mechanism originating from the combined rheological properties of C-PVA gel and P-PVA solution. 

Narita et al. investigated the linear and nonlinear viscoelasticity and fracture behaviors of the DC PVA gels, [7,14,15,16,17,18] and demonstrated that the introduction of transient cross-links significantly improved the fracture toughness. Our group studied the nonlinear stress relaxation of the DC PVA gels under various types of deformation, and showed that the effects of time and strain on the nonlinear viscoelasticity are separable [19]. This simple feature facilitates the establishment of constitutive equations to describe the nonlinear viscoelastic responses. These results indicate that the dual cross-link concept is a promising approach for the toughening of hydrogels.

DC PVA hydrogels have several material parameters that govern the physical properties; the concentrations of elastically effective permanent and physical cross-links, and the concentration and molecular weight of PVA. In particular, balance between the concentrations of permanent and physical cross-links is crucial for controlling the viscoelasticity. According to a protocol in the literature, [7] borate ions were introduced into a C-PVA hydrogel by immersing in an aqueous borax solution. The equilibrium concentration of the borate ions incorporated into the hydrogel (*C*_B_^IN^) is governed by thermodynamic equilibrium; however, naturally, the *C*_B_^IN^ value is not identical to the initial concentration of the borate ions in the external solution (*C*_B0_^EX^), and there is no definite relationship between *C*_B0_^EX^ and *C*_B_^IN^. The concentration *C*_B_^IN^ is an important parameter that regulates the physical properties of DC-PVA gels. However, previously, results were analyzed using *C*_B0_^EX^ as a parameter [18] because the *C*_B_^IN^ values were not determined and the direct determination of *C*_B_^IN^ was difficult. For a comprehensive understanding of the physical properties of DC-PVA gels, quantification of *C*_B_^IN^ and data analysis using the determined *C*_B_^IN^ values should be performed.

We employ the azomethine-H method to determine *C*_B_^IN^. Azomethine-H forms a stable colored complex with boric acid [20,21]. Due to a proportional relationship between the absorbance and the concentration of the complex, the borate ion concentrations of the external solutions before and after the immersion of the gels (*C*_B0_^EX^ and *C*_B__∞_^EX^, respectively) can be evaluated. The concentration *C*_B_^IN^ can be obtained based on the reduction of *C*_B0_^EX^ to *C*_B__∞_^EX^. We compare the linear dynamic viscoelasticity of a DC-PVA gel, the C-PVA gel with the same permanent cross-link concentration (*μ*_c_) as in the DC-PVA gel, and the P-PVA solution with *C*_B_ = *C*_B_^IN^. In particular, by comparing the rheological behavior of the DC-PVA gel and the P-PVA solution with *C*_B_ = *C*_B_^IN^, the effects of permanent cross-linking on the relaxation strength, the relaxation time, and its distribution are examined. We show that the viscoelasticity of the DC-PVA gels cannot be represented by a simple sum of the viscoelasticity of the C-PVA gel with the same *μ*_c_ and the P-PVA solution with the same *C*_B_. We reveal that the relaxation strength in the DC gel is considerably higher than that in the P-PVA solution with the same *C*_B_, whereas the relaxation time in the DC gel is slightly longer but comparable to that in the P-PVA solution. We discuss the effects of the introduction of permanent cross-links into temporary networks on these characteristics.

## 2. Results and Discussion

### 2.1. Borate-Ion Concentration in DC-PVA Gel

A DC-PVA gel DC-88-26 or DC-132-21 was prepared by incorporating borate ions in a PVA gel with chemical cross-links C-88 or C-132, respectively, [7] according to the scheme shown in Figure 2. The gel C-88 or C-132 was immersed in an aqueous solution of borax with the initial borate concentration (*C*_B0_^EX^) of 5.48 mM. A certain amount of salt (NaCl) was mixed with the external solution such that the size of each hydrogel after the introduction of borate ions was equivalent to that in the preparation state of C-88 or C-132. Thus, the complicated effects of volume change on the mechanical properties of hydrogels can be avoided [7]. The external solution after equilibration was employed to determine the borate concentration (*C*_B__∞_^EX^) via the azomethine-H method as *C*_B__∞_^EX^ was required to evaluate *C*_B_^IN^. The details of the gel preparation are provided in the Materials and Methods section.

Figure 3b shows the UV-vis absorbance spectra of the 100-fold diluted external borax solutions containing azomethine-H before and after C-88 immersion. Azomethine-H and boric acid form a stable complex (Figure 3a) [20,21]. The reduction in the finite absorbance of the boron–azomethine-H complex after C-88 immersion reflects the transfer of borate ions from the outside to the inside of the gel. We measured *C*_B0_^EX^ and *C*_B__∞_^EX^ from the absorbance at a wavelength of 410 nm (*A*_410_) using a calibration relationship between *A*_410_ and *C*_B_ [21]. The concentration *C*_B_^IN^ was calculated from the *C*_B0_^EX^ and *C*_B__∞_^EX^ values and the known volumes of the external solution and the gel. The same analysis was performed for DC-132-21.

Table 1 presents the *C*_B_^IN^, *C*_B0_^EX^, and *C*_B__∞_^EX^ values of each DC gel. Importantly, *C*_B_^IN^ for each DC gel is considerably (approximately five times) larger than *C*_B0_^EX^. Note that the difference between *C*_B_^IN^ and *C*_B0_^EX^ depends on not only the volumes of the hydrogel and external solution used in the experiments, but also the material characteristics such as the concentrations of PVA and GA (*C*_PVA_ and *C*_GA_, respectively). The large difference between *C*_B_^IN^ and *C*_B0_^EX^ indicates the significance of determining *C*_B_^IN^ for analyzing the physical properties of DC-PVA gels. The *C*_B_^IN^ values of the two DC gels are not largely different, although *C*_B_^IN^ tends to increase as *C*_GA_ decreases. The degree of equilibrium swelling in the aqueous borax solution (solvent uptake) increases as *C*_GA_ decreases, but as mentioned before, the gel volume after the introduction of borate ion was kept unchanged by adding the salt in water. The isovolumetric condition used here suppresses the effect of *C*_GA_ on *C*_B_^IN^.

### 2.2. Dynamic Viscoelasticity

Figure 4a shows the storage and loss shear moduli (*G′* and *G″*, respectively) as a function of angular frequency (*ω*) for DC-88-26. For comparison, the corresponding data of C-88 with the same *C*_GA_ as in DC-88-26 and no physical cross-links, and P-26 with the same *C*_B_ as in DC-88-26 and no chemical cross-links, are also shown. Figure 4b displays the corresponding data for DC-132-21. The characteristic parameters of each specimen are listed in Table 2. C-88 and C-132 exhibit *ω*-independent *G′* and a relationship of *G′* >> *G″* over the entire *ω* range, which are the features of elastic solids. P-26 and P-21 show a typical viscoelastic liquid behavior, which is characterized by a crossover between *G′* > *G″* at high *ω* and *G′* < *G″* at low *ω*. DC-88-26 and DC-132-21 exhibit a viscoelastic solid behavior, which is characterized by two quasi-plateau regions of *G′* in the high and low *ω* range (*G*_∞_ and *G*_0_, respectively) and a pronounced peak of *G″* at a characteristic frequency (*ω*_c_). Evidently, the viscoelastic solid features of the DC gels arise from the combined characteristics of the chemical and physical gels, as demonstrated earlier [7]. As can be seen in Table 2, the *G*_0_ value of each DC gel was similar to that of the corresponding chemical gel. This agreement indicates that the permanent network governs the modulus in the equilibrium state (*G*_0_) of the DC gels, in which the temporary chains formed via physical cross-links fully relax. According to the classical phantom network model [22], the shear modulus of tetrafunctional polymer networks without structural defects (*G*_ph_) is presented by *G*_ph_ = (1/2)*νRT* = *μRT*, where *ν*, *μ*, *R*, and *T* are the molar concentration of network strands, molar concentration of cross-links, the gas constant, and absolute temperature, respectively. From the *G*_0_ values of C-88 and C-132, the elastically effective chemical cross-link concentrations (*μ*_c_) were evaluated to be 4.1 and 5.5 mM, respectively. Each of the *μ*_c_ values is smaller than *C*_GA_ in the feed (8.8 and 13.2 mM, respectively), indicating the presence of finite structural defects such as elastically inactive dangling and loop chains, which are often observed in conventional gels [22].

The *G′*– and *G″*–*ω* curves of P-21 and P-26 are almost identical because the difference in *C*_B_ is very small. The appreciable effects of *C*_GA_ on the dynamic viscoelasticity are observed in the comparison of the data of DC-88-26 and DC-132-21. As *C*_GA_ increases, the relaxation strength indicated by the difference of the two plateau values of *G′* or the peak height of *G″* increases. In contrast, *C*_GA_ has no appreciable effect on the characteristic frequency *ω*_c_ at the *G″* peak, although *C*_GA_ has a finite effect on the relaxation time distribution represented by the peak width. The effects of *C*_GA_ on these characteristics will be discussed later together with the effect of the introduction of permanent cross-links on the relaxation dynamics.

Figure 5a shows a comparison between the *G′*– and *G″*–*ω* curves of DC-88-26 with the sum of the corresponding data of C-88 and P-26. The *G″–**ω* curve of the sum near *ω*_c_ is almost identical to that of P-26 because of the negligibly small dissipative property of C-88. Clearly, the dynamic viscoelasticity of DC-88-26 cannot be represented by the sum of the dynamic viscoelasticity of P-26 and C-88. DC-88-26 exhibits a higher relaxation strength and slower relaxation than the sum. The same tendency is also observed for DC-132-21, which is shown in Figure 5b. We discuss the dynamic viscoelasticity of these two DC gels in terms of the following three aspects: (i) relaxation strength, (ii) relaxation-time distribution, and (iii) relaxation time.

#### 2.2.1. Relaxation Strength

With respect to (i), the onset of the *G′* plateau in P-26 or P-21 was observed at high *ω* (Figure 4). The *G*_∞_ value represents the contribution of all elastically effective temporary chains formed via physical cross-links in the solution. The *G′* value at 100 s^‒1^ for an aqueous PVA solution of the same concentration (*C*_PVA_ = 12 wt.%) without boron (approximately 200 Pa) is more than one order of magnitude lower than those for P-26 and P-21 (Figure 4c). This confirms that the *G*_∞_ of P-26 and P-21 can be entirely attributed to the contribution of the elastically active temporary chains generated via the interchain physical cross-links formed by tetrahydroxy borate ions. Using the *G*_∞_ (approximately 8.4kPa) of P-26 and P-21, the concentration of the di–diol complex of boron (*μ*_p_)—which forms the interchain bonds—is evaluated to be approximately 3.4 mM with the relation *G*_∞_ = *G*_ph_ = *μ*_p_*RT*. The *μ*_p_/*C*_B_^IN^ value suggests that approximately 13% of the total boron is used in the formation of interchain bonds. In P-PVA solutions, various states, that is, free, dangling bond (mono–diol complex), and intrachain bond (di–diol complex) states, of borate ions exist, other than the interchain bond (di–diol complex) state [23]. The first three of these four states do not contribute to the elasticity. Using ^11^B nuclear magnetic resonance (NMR) spectroscopy, Huang et al. [23]. revealed that 35% of the total boron formed a di–diol complex in a dilute P-PVA solution with a low borax content. The low *μ*_p_/*C*_B_^IN^ value of P-26 and P-21 is plausible considering that a finite amount of the di–diol complex can form intrachain bonds that do not contribute to the modulus and that the concentrations are considerably different.

For the DC gels, the difference between the *G*_∞_ and *G*_0_ values (Δ*G*(DC)) corresponds to the relaxation strength of all viscoelastic chains (Figure 4). For the DC gels, *G*_∞_ represents the contribution of all elastically effective network strands generated via permanent and temporary cross-links, and *G*_0_ denotes the contribution of all elastically effective network strands formed via permanent cross-links. In the case of DC-88-26, Δ*G*(DC) (20.7 kPa) is approximately twice as high as Δ*G*(P-26) (8.5 kPa). The higher relaxation strength of DC-88-26 is also verified by comparing the peak heights of *G″* (*G″*_peak_) for DC-88-26 and P-26, which are a simple measure of the relaxation strength (*G″*_peak_ = 8.1 and 3.0 kPa for DC-88-26 and P-26, respectively). Qualitatively, the same tendency is observed for DC-132-21: Δ*G*(DC) (37.9 kPa) is approximately four times as high as Δ*G*(P-21) (8.20 kPa), and *G″*_peak_ = 12.7 and 3.0 kPa for DC-132-21 and P-21, respectively. The higher relaxation strength, Δ*G*(DC) > Δ*G*(P), implies that each DC gel comprises a significantly larger number of viscoelastic chains with finite relaxation times than the corresponding physical gel.

The various schemes shown in Figure 6 explain how the introduction of permanent cross-links into a temporary network increases the number of viscoelastic chains with finite relaxation times, even when the number of transient cross-links remains unchanged. In scheme (I), the formation of a single permanent cross-link divides two temporary chains with both ends connected to physical cross-links (TCs) into four temporary chains with one end tethered to the permanent cross-link (O-TCs). In scheme (II), two temporary chains with one free end (F-TCs) are split into two O-TCs and two dangling chains with one end tethered to the permanent cross-link and one free end (O-Cs). In scheme (III), a pair of one TC and one F-TC is divided into three O-TCs and one O-C. All schemes result in an increase in the viscoelastic chains with finite relaxation times. Characteristically, each scenario yields two viscoelastic chains per permanent cross-link. This simply means that the number of viscoelastic chains is double the amount of permanent cross-links, thereby increasing the relaxation strength; *ΔG*** = *ΔG*_ph_ = *μ*_c_*RT*, which is equivalent to the *G*_ph_ of a network with only permanent cross-links. When all the viscoelastic chains formed by the introduction of permanent cross-links contribute to *G*_∞_, ***ΔG*****_DC,model_** is presented by
***ΔG*****_DC,model_** = Δ*G*(P) + *ΔG*** + Δ*G* (C)(1)
where Δ*G*(C) represents a modest degree of relaxation strength of the network with only permanent cross-links resulting from the dangling chains and heterogeneous network structures. The values of ***ΔG*****_DC,model_** for DC-88-26 and DC-132-21 are 22.6 and 29.8 kPa, respectively, and they approximately explain the experimental values of Δ*G* = 20.7 kPa for DC-88-26 and 37.9 kPa for DC-132-21 (Table 2). The agreement should be considered satisfactory in view of the simplifying assumptions in this scenario. The agreement validates the schemes shown in Figure 6 to explain the increase in the number of viscoelastic chains with the introduction of permanent cross-links.

#### 2.2.2. Relaxation-Time Distribution

Regarding (ii), Figure 7 shows a comparison between the shapes of the *G″* peaks for DC-88-26, DC-132-21 and P-26. The shape of the relaxation peak reflects the relaxation time distribution. For comparison, *G″* and *ω* are reduced by the maximum value (*G″*_max_) and angular frequency (*ω*_c_) at the peak for each specimen, respectively. The corresponding data of a dilute P-PVA solution of *C*_PVA_ = 1.75 wt% with *C*_B_ = 26 mM are also shown in the figure. The *ω* dependence of G″ of the dilute P-PVA solution is close to the response of the single Maxwell model presented by *G″*(*ω*)/*G″*_max_ = *ωτ*/(1+*ω*^2^*τ*^2^) (dashed line in the figure). Koike et al. [12]. also observed the same tendency for the dilute P-PVA solution *C*_PVA_ = 2.0 wt% and a borax concentration of 30 mM. The broader relaxation time distribution of the P-PVA solution is attributed to the considerably higher *C*_PVA_ (12 wt.%). In fact, the terminal flow behavior of *G″* ~ *ω*^1^ is not observed in the *ω* range examined herein (Figure 4a,b). The shapes of the *G″* curves for the DC gels and P-PVA solutions are similar at *ω* > *ω*_c_, whereas the DC gels exhibit a weaker *ω* dependence of *G″* at *ω* < *ω*_c_ than the P-PVA solutions. The relaxation broadening in the DC gels is ascribed to the widening of the length distribution of viscoelastic chains resulting from the non-uniform spatial distribution of permanent cross-links and the increase in relaxation modes caused by various types of viscoelastic chains (Figure 6). The relaxation broadening in the low *ω* region of *ω* < *ω*_c_ indicates that the effect of the non-uniform cross-link distribution is pronounced in the dynamics in the spatially large-scale.

#### 2.2.3. Relaxation Time

With respect to (iii), the relaxation time (*τ*) values are evaluated from the inverse of *ω*_c_ at the maximum of the *G″*–*ω* curves, that is, *τ* = 1/*ω*_c_. The relaxation dynamics of temporary networks are regulated by the magnitude correlation between the Rouse relaxation time of the chain (*τ*_Rouse_), the breaking time for a transient cross-link to break (*τ*_break_), and the healing time (*τ*_heal_), which is the time required for the process starting from the detachment to the reformation of transient cross-links [18,24]. This correlation is expected to depend intricately on *C*_GA_ and *C*_B_^IN^. In the present conditions, *τ* is simply governed by *τ*_break_, because the *τ* values for the P-PVA solutions and DC gels are comparable to that for a dilute P-PVA solution of *C*_PVA_ = 1.75 wt% with the same *C*_B_, which is shown in Figure 8. The dynamic viscosity (given by *G*″/*ω*) of the dilute P-PVA solution of *C*_PVA_ = 1.75 wt% is about two orders of magnitude lower than that of *C*_PVA_ = 12 wt%, but the *τ* values for the two P-PVA solutions (about 0.1 s) are nearly identical. Furthermore, the *τ* values for the DC gels (about 0.4 s) are almost similar to those for the P-PVA solutions, although they have covalent cross-links. Thus, *τ* is not significantly influenced by *C*_PVA_ and the amount of covalent cross-links in the range examined here, indicating that *τ* is essentially governed by the lifetime of a transient cross-link (di-diol complex of boron). Moreover, it can be seen in Figure 4c that the *τ* value in the dilute PVA solution without transient cross-link is by far less than 0.01 s. The viscoelastic chains in the P-PVA solutions are shortened via physical cross-links as compared to those in the solution without borax. These results signify *τ*_break_ >> *τ*_Rouse_ in the present conditions.

The *τ* values for the DC gels are comparable to but about four times larger than those for the P-PVA solutions with the same *C*_B_. It should be noted again that the introduction of covalent cross-links appreciably broadens the relaxation in the long-time region, as shown in Figure 7. Therefore, the slightly larger *τ* values for the DC gels are mainly attributed to this broadening effect of the relaxation time distribution in the long-time region. These results indicate that the introduction of covalent cross-links does not alter the governing relaxation mechanism dictated by the breaking of transient cross-links, while it affects the relaxation-time distribution in the long-time regime.

It is intriguing to investigate systematically the effects of *C*_B_^IN^ and *C*_GA_ on the dynamic viscoelasticity of DC-PVA gels with determined *C*_B_^IN^ values. The quantification of *C*_B_^IN^ by the azomethine-H method demonstrated herein will enable precise analyses of the viscoelasticity of DC-PVA hydrogels. The corresponding study will be reported in a separate work in the near future.

## 3. Conclusions

The azomethine-H method enabled us to determine the concentrations of incorporated borate ions to form transient cross-links in the DC-PVA hydrogels (*C*_B_^IN^). Comparison between the dynamic viscoelasticity of the DC-PVA gel and the P-PVA solution with the same *C*_B_^IN^ reveals that the DC gel has a higher relaxation strength, broader relaxation time distribution, and a slightly larger but comparable relaxation time. The same tendency was confirmed for the DC-PVA gels with different glutaraldehyde concentrations (*C*_GA_) and similar *C*_B_^IN^. The introduction of permanent cross-links into a temporary network yielded viscoelastic chains in which one end was tethered to a fixed cross-link and the other end was free or connected to a physical cross-link, increasing the relaxation strength of DC gel. The relaxation times for the DC gels are comparable to that of a dilute P-PVA solution with the same *C*_B_, indicating that the relaxation dynamics is governed by the lifetime of a transient cross-link (di-diol complex). The introduction of permanent cross-links does not alter the governing relaxation mechanism, although it broadens the relaxation time distribution in the long time region.

## 4. Materials and Methods

### 4.1. Sample Preparation

The DC-PVA hydrogels (DC-88-26 and DC-132-21) were prepared using a method reported in the literature (Figure 2) [7]. The number-average molecular weight and degree of hydrolysis of PVA (Kuraray, Tokyo, Japan) were 1.06 × 10^5^ and 98.5%, respectively. The C-PVA hydrogels (C-88 and C-132) were synthesized using glutaraldehyde (GA; Sigma-Aldrich, Darmstadt, Germany) as a cross-linker. An aqueous solution with a PVA concentration (*C*_PVA_) of 12 wt.% was mixed with GA (*C*_GA_ = 8.8 or 13.2 mM) at pH = 1.7. The pH was controlled by adding hydrochloric acid (Nakalai Tesque, Kyoto, Japan). The solution was transferred to a mold and allowed to undergo gelation at 25 °C for one day. The resultant gel with the dimensions of 30 × 30 × 2.3 mm was left to swell in water (500 mL) for washing for 2 days, and the water was renewed several times. The swollen C-88 or C-132 was immersed in an aqueous solution (75 mL) of borax (Na₂B₄O_7_·10H₂O; Fujifilm Wako Pure Chem. Corp., Osaka, Japan) for five days. The initial borate concentration of the external solution (*C*_B0_^EX^) was 5.48 mM. The salt NaCl (1.24 and 1.34 g for DC-88-26 and DC-132-21, respectively) (Sigma-Aldrich, Darmstadt, Germany) was added to the external solution such that the gel dimensions after the incorporation of borate ions were the same as those in the preparation state (30 × 30 × 2.3 mm^3^). After equilibration, the hydrogels were employed for mechanical tests, and the external solutions were used to determine the borate concentrations. 

For comparison, we also used the P-PVA solution (P-26 or P-21) with the same *C*_B_ value as in DC-88-26 or DC-132-21, respectively. Each P-PVA solution was prepared by mixing an aqueous PVA solution that had a *C*_PVA_ = 12 wt.% with a desired amount of borax.

### 4.2. Characterization of Borate-Ion Concentration in DC-PVA Gel

The concentrations of the borate ion incorporated in the gels (*C*_B_^IN^) were evaluated based on the variation in the borate concentrations of the external solutions before and after the immersion of the gel (*C*_B0_^EX^ and *C*_B__∞_^EX^, respectively). The azomethine-H method was used to determine *C*_B0_^EX^ and *C*_B__∞_^EX^ [21]. The external aqueous solutions before and after the gel immersion were diluted by a factor of 100. Each diluted solution was mixed with an aqueous solution of azomethine-H [8-hydroxy-1-(salicylideneamino)naphthalene-3,6-disulfonic acid monosodium salt; Tokyo Chemical Industry, Tokyo, Japan] to achieve an azomethine-H concentration of 1.97 mM. The UV–vis spectra of the solutions were measured using a UV-2550 spectrometer (Shimadzu Corp., Kyoto, Japan). The absorbance of the boron–azomethine-H complex at a wavelength of 410 nm (*A*_410_) was used to evaluate *C*_B0_^EX^ and *C*_B__∞_^EX^ using a calibration relationship between *A*_410_ and *C*_B_ [21]. The *A*_410_ and *C*_B_ relationship was separately obtained by measuring *A*_410_ for the aqueous solutions with known borate ion concentrations. The *C*_B_^IN^ value was calculated from the *C*_B0_^EX^ and *C*_B__∞_^EX^ values and the known volumes of the solution and gel.

### 4.3. Dynamic Viscoelasticity Tests

The linear dynamic viscoelasticity of the DC- and C-PVA gels was measured by an RSA-G2 solid analyzer (TA Instruments, New Castle, DE, USA) in the compression mode at 25 °C. The linear strain regime was confirmed by the strain-amplitude sweep measurements, and the strain amplitude in the linear regime (1%) was employed. The three gel sheets were placed on top of one another such that the height of the laminates was 6.9 mm. This compression geometry was advantageous to effectively prevent the evaporation of water during the measurements and to obtain a large force with a high signal-to-noise ratio. The dynamic viscoelastic data agreed with those obtained in the stretching mode with a single gel strip, thereby confirming that this geometry had no negative effect on the data. The measured complex Young’s moduli (*E**) were converted to the complex shear moduli (*G**) using the relation *G** = *E**/3 for incompressible materials.

The complex shear moduli (*G**) of the P-PVA solutions were measured by an oscillatory rheometer (MCR 502, Anton Paar) using a parallel plate geometry with a diameter of 25 mm at 25 °C.

## Figures and Tables

**Figure 1 gels-07-00071-f001:**
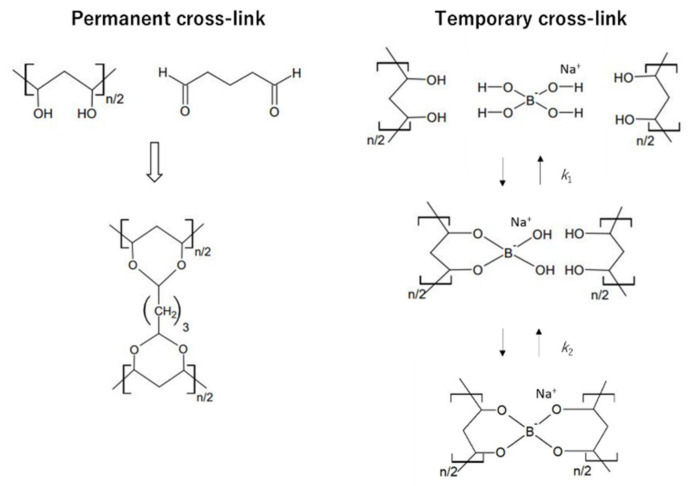
Chemical structures of the permanent cross-link formed by glutaraldehyde and temporary cross-link generated by a tetrahydroxy borate ion.

**Figure 2 gels-07-00071-f002:**
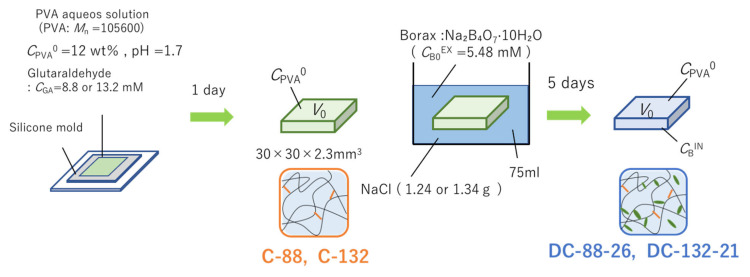
Schematic of the preparation of DC-PVA gel.

**Figure 3 gels-07-00071-f003:**
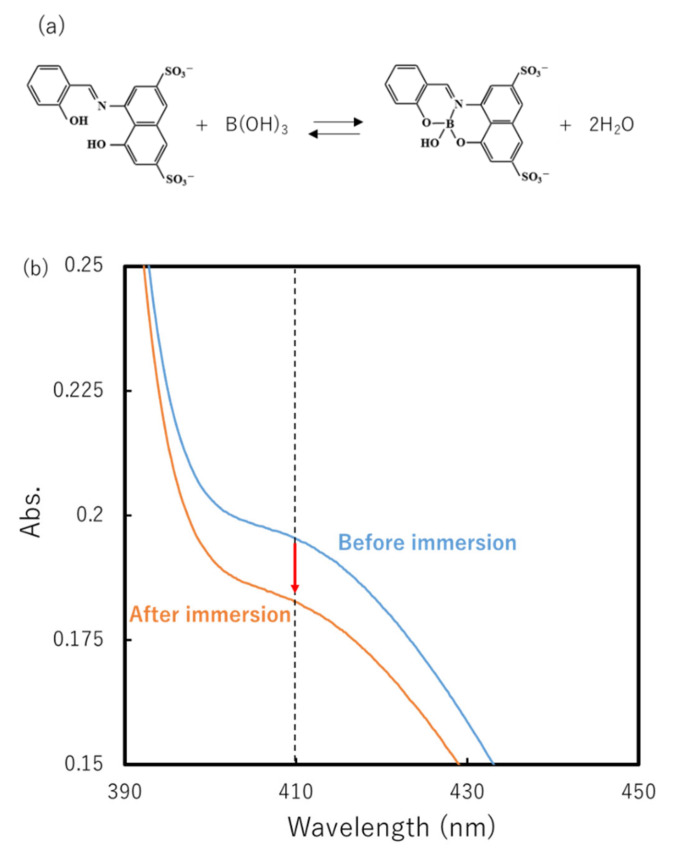
(**a**) Formation of a boron–azomethine-H complex. (**b**) Transmission spectra of the 100-fold-diluted external borax solutions containing azomethine-H before and after C-88 immersion. Absorbance at a wavelength of 410 nm is used for evaluating the borate ion concentration.

**Figure 4 gels-07-00071-f004:**
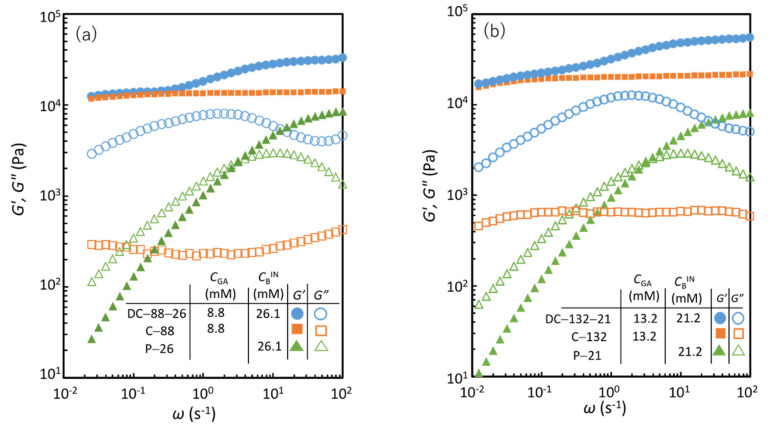
Storage shear modulus (*G′*) and loss shear modulus (*G″*) as a function of angular frequency (*ω*) for (**a**) DC-88-26, C-88, P-26, (**b**) DC-132-21, C-132, P-21, and (**c**) a PVA solution of *C*_PVA_ = 12.0 wt% without borax.

**Figure 5 gels-07-00071-f005:**
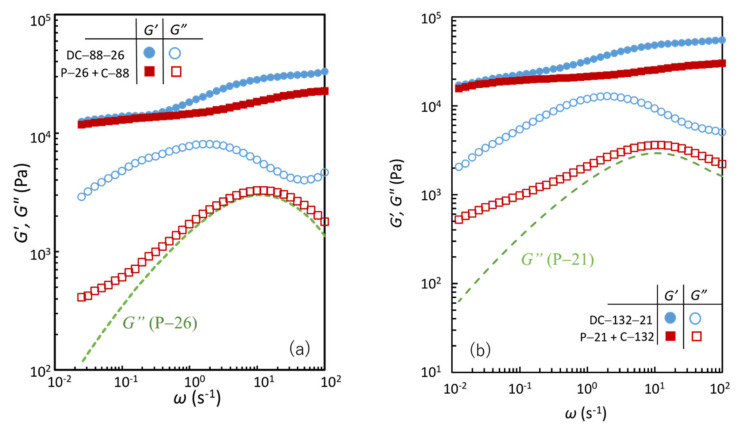
Comparison of the angular frequency (*ω*) dependence of storage shear modulus (*G′*) and loss shear modulus (*G″*) for (**a**) DC-88-26 and the summation of the corresponding data for C-88 and P-26, and (**b**) DC-132-21 and the summation of the corresponding data for C-132 and P-21. The *G″*–*ω* curves of P-26 and P-21 in Figure 4 are shown for comparison.

**Figure 6 gels-07-00071-f006:**
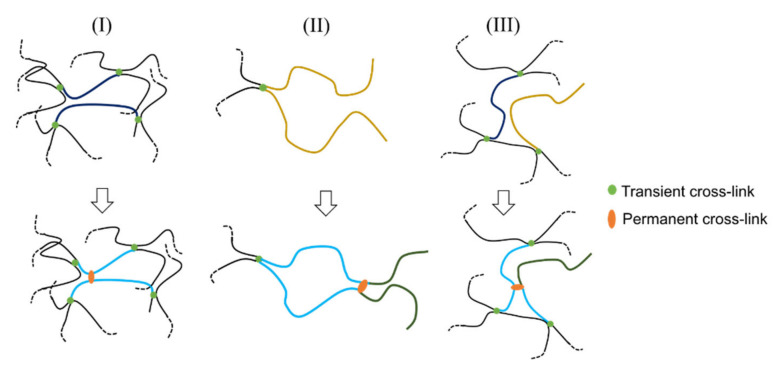
Schemes depicting the increase in the number of viscoelastic chains via the introduction of permanent cross-links.

**Figure 7 gels-07-00071-f007:**
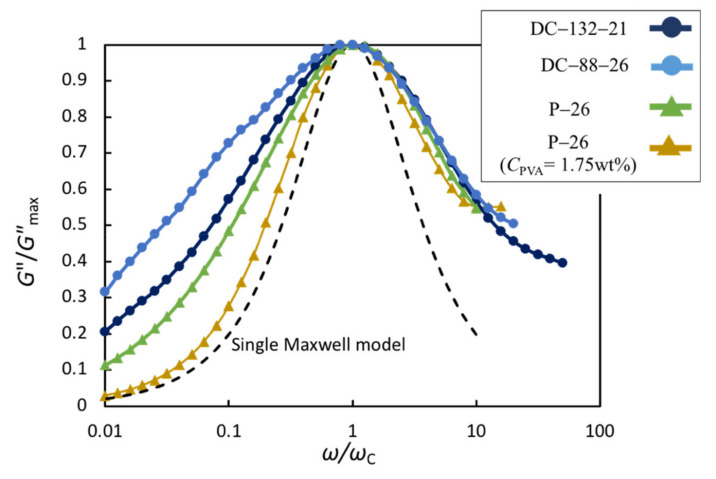
Comparison of the relaxation peaks of *G″* for DC-132-21, DC-88-26, P-26 with *C*_PVA_ = 12.0 wt% and P-26 with *C*_PVA_ = 1.75 wt%. *G″* and *ω* are reduced by *G″*_max_ and *ω*_c_ for each specimen, respectively. The dashed line depicts the corresponding curve of a single Maxwell model.

**Figure 8 gels-07-00071-f008:**
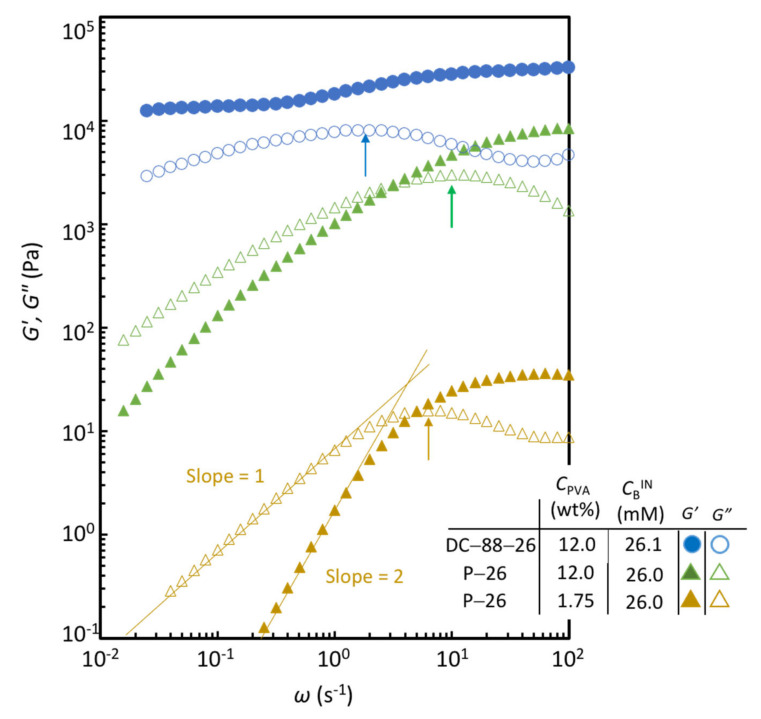
Comparison of the *ω* dependence of *G*′ and *G*″ for DC-88-26, P-26 with *C*_PVA_ = 12.0 wt% and P-26 with *C*_PVA_ = 1.75 wt%.

**Table 1 gels-07-00071-t001:** Borate ion concentrations in the external solutions before and after gel immersion (*C*_B0_^EX^ and *C*_B__∞_^EX^, respectively) and in the DC gels (*C*_B_^IN^).

	*C*_B0_^EX^ (mM)	*C*_B__∞_^EX^ (mM)	*C*_B_^IN^ (mM)
DC-88-26	5.48	4.60	26.1
DC-132-21	5.48	4.84	21.2

**Table 2 gels-07-00071-t002:** Characteristic parameters of specimens.

Specimen	*C*_PVA_ (wt.%)	*C*_GA_ (mM)	*C*_B_^IN^ (mM)	*G*_0_ (kPa) ^a^	*G*_∞_ (kPa) ^b^	*ΔG* (kPa) ^c^	*ΔG*_DC,model_ (kPa)
DC-88-26	12.0	8.80	26.1	12.4	33.1	20.7	22.6
P-26	12.0	0	26.1	0	8.50	8.50	-
C-88	12.0	8.80	0	11.4	14.1	2.70	-
DC-132-21	12.0	13.2	21.2	16.9	54.8	37.9	29.8
P-21	12.0	0	21.2	0	8.20	8.20	-
C-132	12.0	13.2	0	15.4	21.6	6.20	-

^a^*G*′ value at the lowest *ω* in the measurement; ^b^
*G*′ value at *ω* = 100 s^−1^; ^c^ Δ*G* = *G*_∞_ – *G*_0._

## Data Availability

The data presented in this study are available on request from the corresponding author.

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
