# Peer review of "Linear Dynamic Viscoelasticity of Dual Cross-Link Poly(Vinyl Alcohol) Hydrogel with Determined Borate Ion Concentration"

_gels, 2021, doi:10.3390/gels7020071_

Round 1

Reviewer 1 Report

Review comment:

It is known that the strategy to effectively enhance the mechanical strength of hydrogels is to introduce the dynamic bonds to network structure. Understanding the relaxation behavior of dynamic bonds on the mechanical behavior of hydrogels is necessary and interesting. In this work, the authors revisit the classic dual cross-link DC-PVA hydrogels with permanent and dynamic cross-links, and their dynamic viscoelasticity were discussed. Although the dynamic behavior of DC-PVA hydrogels were previously reported by Prof. Narita’s group, comprehensive understanding of their physical behavior in this work is still interesting, including the relaxation strength, relaxation-time distribution, and relaxation time. As a conclusion, this paper is suitable for publication after minor revision. Specific comments are as follows:

(1)Page 5, line 145 and line146, the code aG and bG is inconsistent with the description ( G0 and G∞ ) in Table 2. Please check it.

(2)Figure 8, The interesting phenomena is the asymmetry in the relaxation shape of the G” curves for the DC and P-PVA hydrogels. Could you explain this?

(3) Page 10, line 318. The authors explain that the larger τ value for DC-88-26 hydrogels than that of P-26 hydrogels is ascribed to the reduction of healing time of dynamic bonds by the introduction of permanent crosslinkers. Could you give some evidence to prove this?

Reviewer 2 Report

Taniguchi and Urayama present a manuscript on a dual crosslinked PVA hydrogel. They describe that the combination of permanent and transient cross-links leads to a viscoelastic behavior that cannot be described simply by the sum of the two individual components. I am not sure that this is a surprise at all, but of course a proper rheological characterization is certainly warranted and valuable. I should admit that I am no particular expert on borate ester cross-linked hydrogels, so I am not particularly familiar with the literature. Obviously, and as cited by the authors, some precedent can be found in the literature on essentially the same systems. Accordingly, this contribution can be seen to add some more detail to an already studied system. Here, the authors use a colorimetric test for borate to determine the concentration of borate ions incorporated into the hydrogels, which certainly yields valuable information. Overall, the contribution of the present contribution is valuable, albeit somewhat limited. I am not entirely convinced that this work should be published separately, but maybe better combined with the follow-up paper already suggested by the authors. After all, the experimental basis for this paper is rather thin and a very limited data set is analysed and presented in a somewhat lengthy manner. In this field of view, Figure 3 is not really necessary. It only shows simple UV-vis spectra used for borate quantification. Figure 5 is a repetition of data already shown in Figure 4 and Figure 6 also shows data previously shown but now analysed in a different manner. It seems to me that all these data, including Figure 8 could actually presented in one compound Figure 3 a-g (or maybe 2 figures).

Minor comments:

The authors write: “The CBIN values of the two DC gels are not largely different, but CBIN tends to decrease as CGA increases”. Could the authors maybe speculate on the reason for this observation?

Page 6, line 156. Should it not also read, which are the features of a viscoelastic solids?

The conclusion would be easier to follow if fewer abbreviations were used, in particular for some readers who might only or first read the conclusion.

Abstract: check phrase: “and the relaxation of via temporary cross links.”

Introduction: The authors write that “Most hydrogels […] are elastic and brittle” That´s a strange characterization, given that the to characteristics are essentially the opposite of each other. The term brittle implies hardness together with a lack of elasticity or flexibility or toughness.

The authors use the capital letter C for concentration. Normally, one uses the small letter.

Page 8, line 254: check syntax “represents the a modest degree”

Figure 4: why did the authors arrange the Figures a on the right and b on the left?

In the materials and methods, please specify the volume of water used to swell/wash the hydrogels and how often this was exchanged. Please specify the “certain amount” of NaCl added.

Please specify how the azomethine-H assay was calibrated.

How did the authors confirm that the amplitude  in the oscillatory measurement was in the range of the linear viscoelastic regime?

Round 2

Reviewer 2 Report

The reviewers answered and addressed all points raised by the reviewer, with the exception of the most important point, that this study is very incremental, based on a very thin experimental basis and does not really add much new knowledge. I still don´t really see why this study should be published at this point, but maybe I am too harsh in this.